# Cerebellar Transcranial Magnetic Stimulation Reduces the Silent Period on Hand Muscle Electromyography During Force Control

**DOI:** 10.3390/brainsci10020063

**Published:** 2020-01-24

**Authors:** Akiyoshi Matsugi, Shinya Douchi, Kodai Suzuki, Kosuke Oku, Nobuhiko Mori, Hiroaki Tanaka, Satoru Nishishita, Kyota Bando, Yutaka Kikuchi, Yohei Okada

**Affiliations:** 1Faculty of Rehabilitation, Shijonawate Gakuen University, Hojo 5-11-10, Daitou city, Osaka 574-0011, Japan; 2Department of Rehabilitation, National Hospital Organization Kyoto Medical Center, Hukakusamukaihatacyo1-1, Husimi-ku Kyoto City, Kyoto 612-8555, Japan; 3Division of Rehabilitation, Hanna Central Hospital, Ikoma, Nara 630-0243, Japan; 4Department of Neuromodulation and Neurosurgery, Osaka University Graduate School of Medicine, Osaka 565-0871, Japan; 5Department of Neurosurgery, Osaka University Graduate School of Medicine, Osaka 565-0871, Japan; 6Graduate School of Health Sciences, Kio University, 4-2-2 Umami-naka, Koryo-cho, Kitakatsuragi-gun, Nara 635-0832, Japan; 7Department of Rehabilitation, Baba memorial Hospital, Nishiku Hamaderahunaotyohigashi 4-244, Sakai City, Osaka 592-8555, Japan; 8Institute of Rehabilitation Science, Tokuyukai medical corporation, 3-11-1 Sakuranocho, Toyonaka City, Osaka 560-0054, Japan; 9Kansai Rehabilitation Hospital, 3-11-1 Sakuranocho, Toyonaka City, Osaka 560-0054, Japan; 10National Center Hospital, National Center of Neurology and Psychiatry, Kodaira 187-0031, Japan; 11Department of Rehabilitation for Intractable Neurological Disorders, Institute of Brain and Blood Vessels Mihara Memorial Hospital, Ohtamachi366, Isesaki City, Gunma 372-0006, Japan; 12Neurorehabilitation Research Center of Kio University, Koryo-cho, Kitakatsuragi-gun, Nara 635-0832, Japan

**Keywords:** cortical silent period, GABAergic circuit, cerebellum, transcranial magnetic stimulation

## Abstract

This study aimed to investigate whether cerebellar transcranial magnetic stimulation (C-TMS) affected the cortical silent period (cSP) induced by TMS over the primary motor cortex (M1) and the effect of interstimulus interval (ISI) on cerebellar conditioning and TMS to the left M1 (M1-TMS). Fourteen healthy adult participants were instructed to control the abduction force of the right index finger to 20% of the maximum voluntary contraction. M1-TMS was delivered during this to induce cSP on electromyograph of the right first dorsal interosseous muscle. TMS over the right cerebellum (C-TMS) was conducted prior to M1-TMS. In the first experiment, M1-TMS intensity was set to 1 or 1.3 × resting motor threshold (rMT) with 20-ms ISI. In the second experiment, the intensity was set to 1 × rMT with ISI of 0, 10, 20, 30, 40, 50, 60, 70, or 80 ms, and no-C-TMS trials were inserted. In results, cSP was significantly shorter in 1 × rMT condition than in 1.3 × rMT by C-TMS, and cSP was significantly shorter for ISI of 20–40 ms than for the no-C-TMS condition. Further, motor evoked potential for ISI40-60 ms were significantly reduced than that for ISI0. Thus, C-TMS may reduce cSP induced by M1-TMS with ISI of 20–40 ms.

## 1. Introduction

Transcranial magnetic stimulation (TMS) is often used for assessing motor control associated with the primary motor cortex (M1) [1]. Repetitive [2,3], paired-pulse [4], and single-pulse TMS enables the probing of neural circuit function in M1 [1]. Single-pulse TMS induces a silent period (SP) on electromyography (EMG) in hand muscles during the slight voluntary muscle contraction that follows motor-evoked potentials (MEPs) [5,6]. The cortical SP (cSP) induced by TMS over M1 is mediated by cortical inhibitory neural circuits [6,7], and the cortical inhibitory function is likely mediated by gamma-aminobutyric acid (GABA) in the human brain [6,8]. In our previous study, modulation of cSP duration was dependent on the actual output force and EMG activity for force control [9], indicating that inhibitory GABAergic neural circuits estimated by cSP partially contributed to force control. Another study reported that cSP in patients with cerebellar ataxia was abnormal [10,11] especially in those with dysfunction of dentate thalamic tracts estimated with paired-pulse TMS testing [11], indicating the cSP may be associated with cerebellar function. However, there is a lack of evidence that GABAergic neural circuits estimated by cSP are associated with the cerebellum.

Paired-pulse TMS is often used to estimate cerebellar effects on the contralateral M1 [11,12,13,14]. Using this method, cerebellar stimulation is applied approximately 8 ms prior to test TMS over the contralateral M1, and the resulting test MEP is reduced [11,14]. An electroencephalography (EEG) study reported that single-pulse TMS over the cerebellar hemisphere activated the contralateral M1 from 20 to 40 ms after stimulation [15,16,17]. Therefore, there may be effects on M1 from 20 to 40 ms after single-pulse cerebellar TMS. Based on these hypotheses, in this study, we investigated whether cerebellar TMS would modulate cSP after M1-TMS beyond this time window, with inter-stimulus intervals (ISIs) of 0, 10, 20, 30, 40, 50, 60, 70, and 80 ms.

In paired-pulse TMS, the conditioning TMS effects depend on the test TMS intensity [18], and the cSP is modulated by the test TMS intensity to M1 [19]. Therefore, the cSP modulation induced by cerebellar conditioning TMS may be affected by the intensity of test M1-TMS. In the first experiment, we aimed to investigate whether the intensity of test TMS over M1 affected the modulation of cSP by cerebellar conditioning TMS, and in the second experiment, we investigated the effect of ISI on cerebellar conditioning and M1-TMS.

## 2. Materials and Methods

### 2.1. Participants

Fourteen healthy volunteers (eight males, mean age ± standard deviation (SD): 19.5 ± 0.5 years) participated in the first and second experiments. All participants were right-handed and had no history of epilepsy or other neurological diseases including depression. There was no participant on medication for neurological/psychiatric diseases. After an explanation of the experimental protocol, informed consent was obtained from all participants for the first and second experiments.

The ethics committee of Shijonawate Gakuen University approved the experimental procedures (ethical approval code: 30-3). This study was conducted according to the principles and guidelines of the Declaration of Helsinki. After an explanation of the experimental protocol, informed consent was obtained from all participants included in the study.

### 2.2. EMG Recordings

To record EMG signals, 2 Ag/AgCl surface-recording electrodes were placed 2 cm apart on the right first dorsal interosseous (FDI) muscle and a reference electrode was placed on the right dorsal wrist (see Figure 1). EMG signals were amplified via an amplifier (MEG-1200, Nihon Kohden, Tokyo, Japan) with a pass-band filter of 15 Hz to 3 kHz. The EMG signals were converted to digital signals at a sampling rate of 10 kHz using an A/D converter (PowerLab 800S, ADInstruments, Colorado Springs, CO, USA), and the digital signals were stored on a personal computer. This recording method was the same as that reported in our previous study [9].

### 2.3. TMS over M1 (as Test Stimulation)

A single monophasic TMS pulse was delivered to M1 by a figure-of-8 coil (YM-132B, Nihon Kohden, Tokyo, Japan) with an outer diameter of 99 mm connected to a magnetic stimulator (SMN-1200, Nihon Kohden, Tokyo, Japan). The center of junction of the figure-of-8 coil was set at the hotspot in the left M1 to induce MEPs in the right FDI. The hotspot was defined as the site where the largest MEP was obtained in the left hemisphere on the head of the participant. This hotspot was marked on the swimming cap the participant was wearing to prevent change in the stimulation site. The current in the coil was directed from anterior to posterior, inducing a PA traveling current in the brain [20,21], and this current stimulation preferentially recruits the late I3-wave [22]. Cerebellar TMS reduced electromyographical responses to I3 waves at an ISI of 6 ms, whereas it did not affect responses to I1 waves [23]. This finding suggests that this cerebellar brain inhibition (CBI) is caused by suppression of I3 waves by cerebellar stimulation. Based on these findings, we thought that the PA direction current was suitable for observing the cerebellar TMS effect. The resting motor threshold (rMT) of the FDI muscle was defined as the minimal intensity of the magnetic stimulator output that produced MEPs with an amplitude larger than 50 µV in at least three out of five stimulations delivered over the hotspot [9,12,24]. The stimulus intensity was set at 1.3 (only in the first examination) or 1.0× the rMT.

### 2.4. TMS over Cerebellum (as Conditioning Stimulation)

Conditioning single-pulse cerebellar TMS was delivered using a double cone coil (YM-133B, Nihon Kohden, Tokyo) with an outer diameter of 60 mm connected to a magnetic stimulator (SMN-1200, Nihon Kohden). To stimulate the right cerebellar hemisphere, the center of the junction of the coil was placed at a site 1 cm below and 3 cm to the right of the inion [12,25]. The current in the coil was directed downward, inducing an upward current in the brain [13]. The intensity of the cerebellar TMS was set at 90% of the rMT of cervicomedullary MEP in the right FDI muscle [14]. The rMT for cerebellar TMS was defined as the lowest stimulation intensity producing a short-latency EMG response, which is cervicomedullary MEP, in the right FDI muscle immediately after the cerebellar TMS in five out of ten consecutive stimuli [14]. If the short-latency EMG response was not evoked even at the maximum stimulator output, the conditioning cerebellar TMS intensity was set at 90% of the maximum stimulator output [12,14,26].

### 2.5. Procedure

The first and second experiments were conducted on the same day separated by an interval of more than 30 min to prevent fatigue. The same 14 participants participated in both the first and second experiments. Figure 1 shows the experimental set up in the first and second experiments. Subjects sat on a chair with a backrest in front of a computer monitor. Their right forearm and hand were fixed to a metal frame to prevent unwanted movements by test M1-TMS or conditioning cerebellar TMS, as reported previously [9]. The abduction force of the right index finger was measured using a force transducer, and participants were instructed to relax all other hand and arm muscles, as reported previously [9]. Before the experiment, the force level of maximum voluntary abduction of the index finger with FDI contraction (MVC) was measured, and the target level was adjusted to 20% of the MVC [9]. The force level output was displayed on the monitor in front of the subject, and the subject was instructed to adjust the force level to 20% MVC target levels. 

In the first experiment, to investigate the appropriate intensity to observe the effect of cerebellar TMS on cSP induced by TMS over the contralateral M1, the cSPs induced by TMS with intensity of 1.0× and 1.3× rMT with and without conditioning cerebellar TMS were measured. These tests (rMT: 1.0/1.3×; cerebellar TMS: on/off) were conducted 10 times in each of the four conditions in random order. The ISI was fixed to 20 ms because the cSP in this ISI was determined in our pilot study. The inter-trial intervals were set to longer than 10 s.

In the second experiment, to test the effects of ISI of M1 and cerebellar TMS, TMS over M1 was delivered during adjustment of the force target, and cerebellar TMS was delivered before the M1-TMS. The ISIs were set at 0, 10, 20, 30, 40, 50, 60, 70, and 80 ms, because the time was twice as long as 40 ms [15] which may be modulated. Ten trials with cerebellar TMS were conducted for each ISI in random order. Ten no-conditioning trials (without cerebellar TMS) were inserted in random order. The inter-trial intervals were set to longer than 10 s.

### 2.6. Data Analysis

Figure 2 shows the analysis of force, background EMG (bEMG), MEParea, and cSP duration for the first and second experiments. The average force level 100–150 ms before M1-TMS was measured in all trials and converted to /MVC. All EMG traces were rectified, and the pre-stimulus average bEMG amplitude was calculated 100–150 ms before M1-TMS in all trials because there were conditioning cerebellar TMS artifacts 10–80 ms before M1-TMS. This obtained bEMG amplitude was converted to /MVC. The MEP area was calculated as the root mean square of EMG activity between MEP onset and offset using the same method with a previous study [27]. The cSP duration was defined as the interval between the TMS onset and cSP offset [6,9,28] (Figure 2). The offset of the cSP was defined as > mean + 3 × SD of the bEMG level in the resting state [9,28], but if EMG baseline drifted, the offset was visually estimated. Therefore, two independent assessors measured cSP, and the average of two cSP values was used for subsequent analysis. Further, the intraclass correlation coefficient (ICC) was calculated to estimate the absolute agreement (by ICC 2,1) and the consistency (by ICC 3,1) in the cSP measured by two independent assessors [29] The mean of Force/MVC, bEMG/MVC, MEParea (mV*ms), and cSP (ms) in each condition were calculated as individual representative values in each condition.

Before analysis of variance (ANOVA), D’Agostino test (skewness and kurtosis) and Kolmogorov–Smirnov (KS) test for normality were conducted to test the normal distribution of data. If data were normally distributed in all conditions, parametric two-way ANOVA in the first experiment (intensity × cTMS) and parametric one-way ANOVA in the second experiment (between no-condition and 0-80ISIs) were conducted. If data were not normally distributed, nonparametric paired t-test (Wilcoxon signed-rank test) in the first experiment, and non-parametric paired one-way ANOVA (Friedman’s test) in the second experiment were conducted. These analyses were applied to the values of actual force level (/MVC), bEMG (/MVC), and cSP duration (ms). To compare the effect of stimulus intensity on cSP duration, conditioned/no-conditioned cSP was calculated and paired t-test was conducted. In the second experiment, when statistically significant differences were found, post-hoc multiple comparison tests (Scheffe’s test) were conducted to test whether there were differences between these nine ISI conditions. The alpha level was set at 0.05 for all analyses. All statistical analyses mentioned above were conducted using the software package Ekuseru–Toukei 2012 (Social Survey Research Information Co., Tokyo, Japan). Further, the power (1 – beta error probability) was calculated for the ANOVA after statistical analysis using G*power [30].

## 3. Results

All participants completed both examinations. There was no participant with appearance of cervicomedullary MEP by cerebellar TMS with 100% MSO in either experiment and this result is in line with previously reported findings [12,31,32]. Therefore, 90% MSO was applied in all examinations.

Regarding the cSP measurement, in the first experiment, ICC (2,1) and ICC (3,1) in cSP were 0.99 and 0.99, respectively. In the second experiment, ICC (2,1) and ICC (3,1) were 0.93 and 0.94, respectively. These findings indicate that there was absolute agreement and consistency between the assessors [29]. A non-normal distribution could be observed in at least one of the parameters to be compared in the first and second experiments (see Table 1 and Table 2). Therefore, non-parametric tests were conducted in both experiments.

In the first experiment, Wilcoxon-signed rank sum test revealed that there was no significant difference between no-conditioned and conditioned value of force in 1rMT (z = 0.283, *p* = 0.78) and 1.3rMT (z = 0.35, *p* = 0.73) (Figure 3A), and bEMG in 1rMT (z = 1.73, *p* = 0.08) and 1.3rMT (z = 0.6, *p* = 0.55) (Figure 3B). Similarly, there was no significant difference between no-condition and conditioned value of MEP area in 1rMT (z = 1.9, *p* = 0.06) and in 1.3rMT (z = 0.53, *p* = 0.59). On the other hand, the MEParea in 1.0 × rMT was significantly lower than that in 1.3 × rMT (z = 3.3, *p* = 0.001) (Figure 3C). However, there was a significant difference in cSP duration between non-conditioned and conditioned trials in 1rMT (z = 3.1, *p* = 0.002) and in 1.3rMT (z = 2.48, *p* = 0.013) (Figure 3D). Furthermore, there was a significant difference in no-conditioned trial of 1rMT and 1.3rMT (z = 3.3, *p* = 0.001). Conditioned/no-conditioned cSP in 1rMT was significantly lower than that in 1.3rMT (z = 2.4, *p* = 0.019) (Figure 3E).

Figure 4 shows a sample waveform of averaged EMG in the second experiment. The Friedman test revealed that there was no significant difference in force (chi-square value = 7.4, d.f. = 9, *p* = 0.6) (Figure 5A) and bEMG (chi-square value= 15.9, d.f. = 9, *p* = 0.07) (Figure 5B), but there was a significant difference in MEParea (chi-square value = 42.6, d.f. = 9, *p* = 0.000003) (Figure 5C) and in cSP duration (chi-square value = 69.6, d.f. = 9, *p* = 0.00000000006) (Figure 5D). The post hoc test revealed that the MEParea for ISI40, ISI50, and ISI60 were significantly lower than that for ISI0 (*p* < 0.05) (Figure 5C), and cSP duration for ISI20, ISI30, and ISI40 were significantly lower than that for no-conditioning and ISI0 (*p* < 0.05) (Figure 5D). The effect size (f = 6.1) was calculated from the mean of cSP in each ISI condition using G*power. Statistical test “ANOVA: Repeated measures, between factors” and “Post hoc: Compute achieved power” result power (1-beta error probability) = 1, indicating the power was sufficient for analyzing the number of subjects in this study. 

## 4. Discussion

The aim of this study was to investigate whether cerebellar TMS modulated the cSP induced by TMS to the contralateral M1. The cSP duration was significantly reduced by cerebellar TMS with 1.0× rMT compared to that with 1.3× rMT, and the cSP duration for ISI of 20–40 ms was significantly reduced compared to that for no-conditioning trials. These findings suggested that the cerebellar stimulation modulated the cSP induced by TMS over the contralateral M1.

In first examination, we confirmed the effect of M1-TMS intensity on cSP-modulation by cerebellar TMS. The cSP in the 1.0× rMT condition was significantly shortened than that in the 1.3× rMT condition. This result was consistent previous findings where the cSP was prolonged by increasing the intensity of TMS over M1 [28,33,34]. Conversely, the amount of cSP-reduction in the 1.0× rMT condition was significantly larger than that in the 1.3× rMT condition by cerebellar TMS, indicating that the effect of cerebellar TMS on cSP may be affected by the intensity of TMS over M1. By increasing the intensity of TMS to M1, the number of interneurons facilitating the corticospinal tract activated by TMS increases [35]. The cSP duration reflects inhibitory GABAergic tone in M1 [6,8,36]; therefore, cSP shortening by cerebellar TMS can indicate decreasing GABAergic tone in M1. It is possible that minor increase in the excitability of inhibitory circuits by cerebellar TMS can be masked by large increase of the facilitatory effect of M1-TMS. Based on these findings, it may be preferable using a TMS intensity of 1.0× rMT to M1 to more clearly obtain the effect of cerebellar TMS on cSP.

It is also possible that cSP modulation was affected by the MEP size [28]. As a result, the MEP area at ISI 40–60 ms was significantly reduced, whereas cSP at ISI 20–40 ms was significantly reduced. If cSP reduction completely depended on the size of the MEP, MEP reduction would have appeared at ISI 20–40 ms. Therefore, we believe that cSP reduction by cerebellar TMS may be nearly entirely independent from MEP modulation. However, we cannot completely exclude the possibility that cSP modulation depends on the MEP size at ISI 40 ms because the cSP and MEP areas were significantly reduced by cerebellar TMS only at that time.

The cSP was significantly reduced only for ISI of 20–40 ms in our study. In a previous EEG study, one side of the frontal lobe was activated 20–40 ms before single-pulse TMS over the contralateral cerebellar hemisphere [15,16,17]. Since the duration of change was the same in both studies, similar modulation of cortical excitability may have been observed. Our results revealed that the single pulse cerebellar TMS may affect cSP for approximately 20 ms, and several hypotheses for this exist. The cerebellum projects not only directly to M1 mediated by the dentato-thalamo-cortical pathway [37] but also multiple sites in the prefrontal lobe [38,39], and the prefrontal lobe projects to M1 [40]. Further, the cerebellar hemisphere has functional projections to the ipsilateral frontal lobe [41], the frontal lobe projects to the contralateral M1 [42], and the cerebellum has functional loops with the basal ganglia [43] which project to M1 [44]. Additionally, another important possible loop from the cerebellum to the cerebellum could be mediated by the cortico-ponto-cerebellar pathway and the cortico-rubro-olivo-cerebellar pathway [45,46]. Therefore, there are multiple pathways by which cerebellar TMS could affect GABAergic inhibitory neural circuits mediated via indirect pathways to M1.

The reasons for the cSP-reduction by cerebellar TMS must be carefully considered. The cSP in cerebellar ataxia patients with significantly reduced cerebellar volume and dentate nuclei and without cerebellar brain inhibition (CBI), is prolonged [10,11], indicating that the cerebellar output may have facilitatory effects on GABAergic inhibitory neural circuits in the contralateral M1. In this study, the cerebellar conditioning TMS reduced the cSP. Cerebellar TMS activates cerebellar output [47]. Based on these notions, we had one hypothesis that TMS would activate the cerebellar output to the GABAergic inhibitory neural circuits in the contralateral M1, resulting in cSP reduction. On the other hand, the CBI was absent in patients with dystonia [48], further another previous study report the cSP was prolonged in dystonia [49]. These findings indicate the possibility that the cerebellar output may inhibit the inhibitory GABAergic circuit resulting in cSP prolongation in dystonia patient. These inconsistent findings, that the length of cSP may be reversed even in patients without CBI, from clinical populations complicate drawing a simple conclusion scientifically. Therefore, the conclusions on the effects of the cerebellar output on the inhibitory GABAergic circuit in the contralateral M1 need to be considered carefully and further research is awaited.

There are some methodological considerations that should be considered. First, pre-stimulus background EMG activity may have affected cSP [9]. However, in this study, there was no significant difference in the bEMG between stimulation conditions. Second, the fatigue of contraction may have affect cSP duration [50,51]. In this study, to prevent the effects of fatigue, the interval between experiments was long, and the target level of force was set to 20% of MVC. Further, to minimize the effects of fatigue, the test conditions was set in random order. Therefore, the effects of fatigue on cSP duration were likely to be small.

The cSP is often measured to investigate reasons for motor dysfunction in stroke-induced palsy [6,52], dystonia [49], and cerebellar ataxia [10,11]. However, especially in cerebellar ataxia, the interpretation of abnormal cSP duration is limited, because cerebellar effects on GABAergic inhibitory neural circuits in M1 estimated with cSP are unclear. Our results of reduced cSP after cerebellar TMS may promote understanding of abnormal cSP in cerebellar ataxia. Further studies are needed to examine whether cSP in spinocerebellar degeneration reflects the degree of deficits in force control, and whether cSP correlates with the degree of ataxia.

## 5. Conclusions

The cSP by M1-TMS was significantly reduced 20–40 ms after single-pulse TMS over the contralateral cerebellar hemisphere during force control. These findings indicate that GABAergic inhibitory neural circuits in M1 can be affected by cerebellar output.

## Figures and Tables

**Figure 1 brainsci-10-00063-f001:**
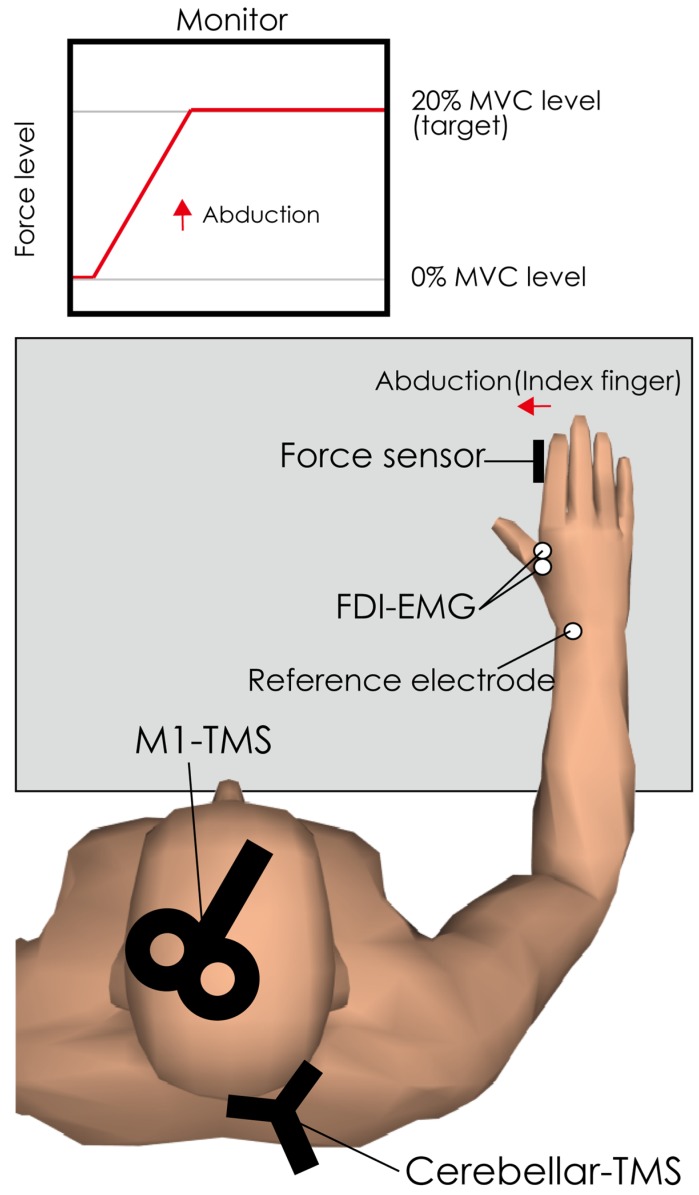
Experimental setup for first and second experiment. The subject was seated in front of a computer monitor. The right hand was fixed with metal frames, and a force sensor was set on the abduction side of the index finger to record the actual output force of abduction of the index finger. The feedback of the actual output force level was provided visually. The subject was instructed to set the force level to the target level (20% of maximum voluntary contraction) with abduction of the index finger. The crossover region of the figure-of-8 coil was set over the hotspot of the right first dorsal interosseous in the left primary motor cortex. The center of the region of double-cone coil was set on site 1 cm under and 3 cm to the right of the inion. FDI; first dorsal interosseous, EMG; electromyography, MVC; maximum voluntary contraction, M1; primary motor cortex.

**Figure 2 brainsci-10-00063-f002:**
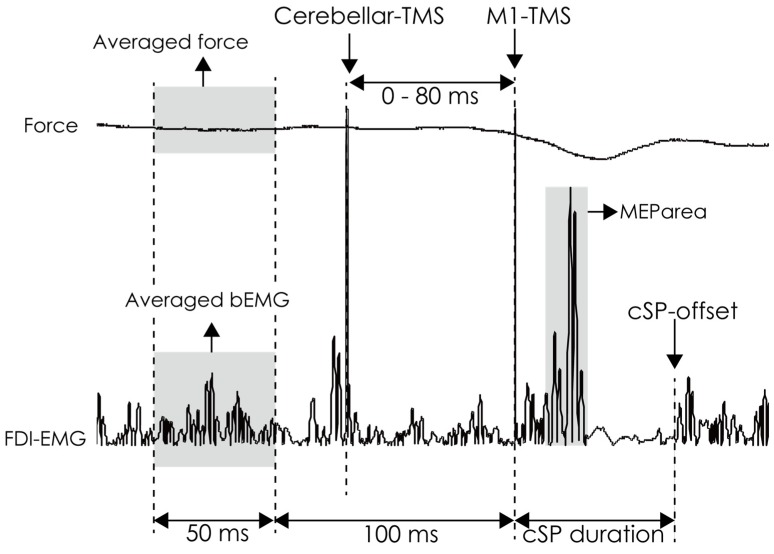
The analysis methods of actual output force and rectified EMG trace. The average force was calculated as the force data 100–150 ms before M1-TMS. The average bEMG amplitude was calculated from the rectified EMG 100–150 ms before M1-TMS. The cSP duration was defined as the time from TMS onset to the time of reappearance of the EMG burst. M1, primary motor cortex; cSP, cortical silent period; EMG, electromyography; bEMG, background electromyography; TMS, transcranial magnetic stimulation.

**Figure 3 brainsci-10-00063-f003:**
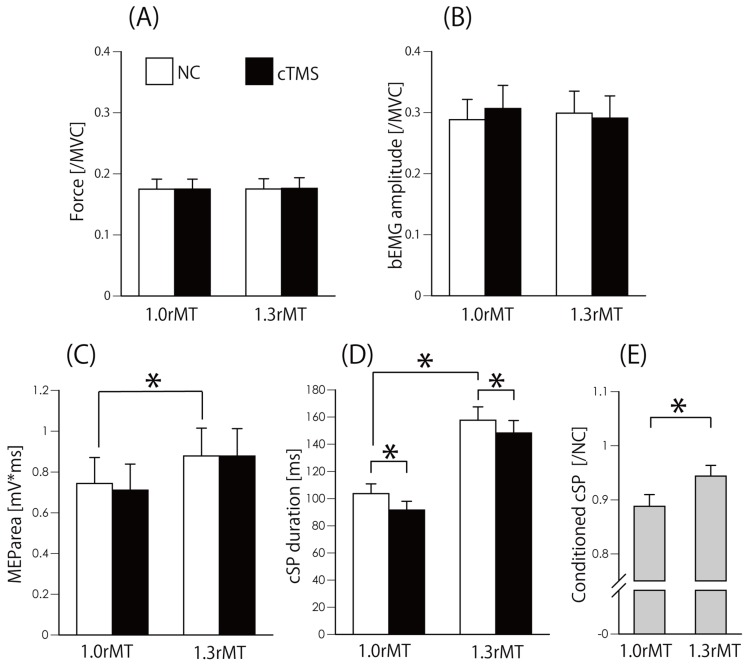
Actual output force (/MVC) (**A**), bEMG amplitude (/MVC) (**B**), MEParea (mV*ms) (**C**), cSP duration (ms) (**D**), and cSP duration (conditioned/no-conditioned) (**E**). Bars indicate the mean, and error bars indicate the standard error. * *p* < 0.05. MVC, maximum voluntary contraction; bEMG, background electromyography; rMT, resting motor threshold; MEP, motor evoked potential; cSP, cortical silent period; NC, no-conditioned; cTMS, cerebellar transcranial magnetic stimulation.

**Figure 4 brainsci-10-00063-f004:**
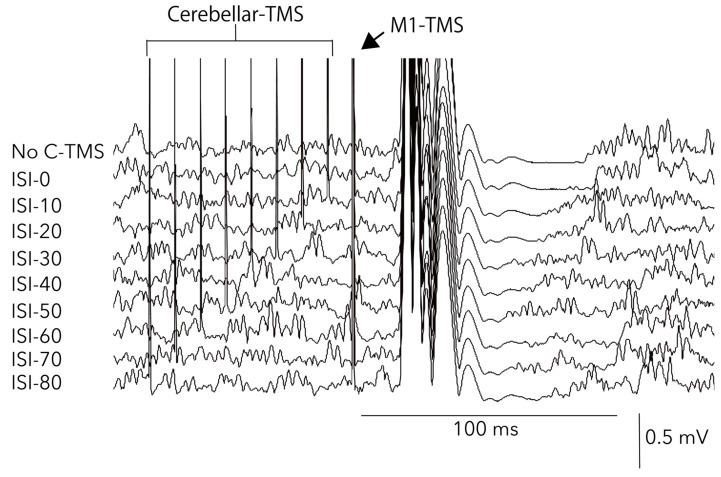
Sample averaged EMG waveforms in the second experiment. ISI, interstimulus interval; C-TMS, cerebellar transcranial magnetic stimulation; M1, primary motor cortex.

**Figure 5 brainsci-10-00063-f005:**
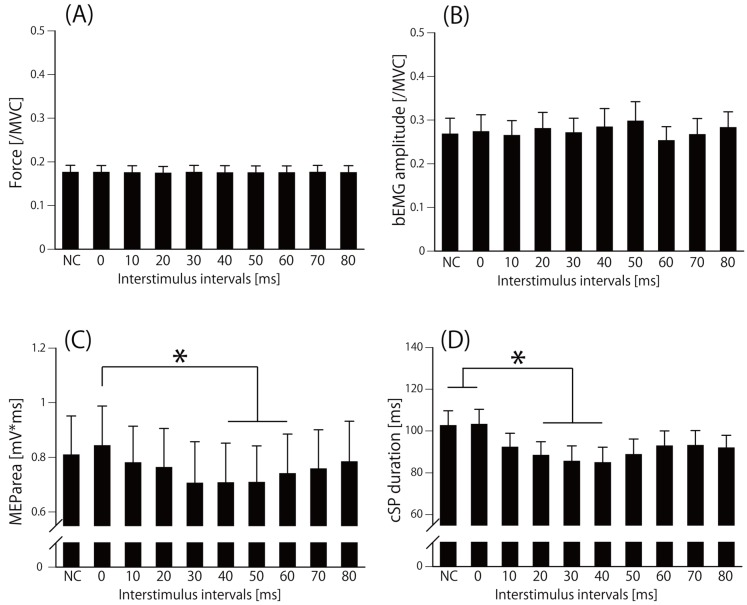
Actual output force (/MVC) (**A**), bEMG amplitude (/MVC) (**B**), MEP area (mV*ms) (**C**), and cSP duration (ms) (**D**) in the second experiment. Bars indicate the mean, and error bars indicate the standard error. **p* < 0.05. bEMG, background electromyography; NC, not conditioned; MEP, motor evoked potential; cSP, cortical silent period.

**Table 1 brainsci-10-00063-t001:** Mean value and D’Agostino and Kolmogorov–Smirnov tests in the first experiment.

			Force [/MVC]				bEMG [/MVC]				MEParea [mV*ms]				cSP [ms]				cSP [C/NC]	
			1rMT_Test	1rMT_cTMS	1.3rMT_Test	1.3rMT_cTMS	1rMT_Test	1rMT_cTMS	1.3rMT_Test	1.3rMT_cTMS	1rMT_Test	1rMT_cTMS	1.3rMT_Test	1.3rMT_cTMS	1rMT_Test	1rMT_cTMS	1.3rMT_Test	1.3rMT_cTMS	1rMT	1.3rMT
Mean			0.17	0.18	0.18	0.18	0.29	0.31	0.30	0.29	0.74	0.71	0.88	0.88	103.7	91.6	157.6	148.2	0.89	0.94
SEM			0.02	0.02	0.02	0.02	0.03	0.04	0.04	0.04	0.13	0.13	0.14	0.13	7.2	6.4	9.8	9.1	0.02	0.02
D’Agostino test	Skewness	Statistic(Z)	2.4096	2.3962	2.4736	2.5681	0.9841	0.7316	0.6486	0.7409	1.4283	1.5369	1.2183	0.9032	2.2252	2.6525	1.6645	1.2351	0.9640	1.8655
		*p*	0.0160	0.0166	0.0134	0.0102	0.3250	0.4644	0.5166	0.4587	0.1532	0.1243	0.2231	0.3664	0.0261	0.0080	0.0960	0.2168	0.3351	0.0621
	Kurtosis	Statistic(Z)	17.8808	15.9005	N.C.	N.C.	6.0121	5.7291	5.8655	5.8438	C.N.C.	C.N.C.	C.N.C.	C.N.C.	C.N.C.	C.N.C.	C.N.C.	C.N.C.	12.7921	C.N.C.
		*p*	0.0000	0.0000	N.C.	N.C.	0.0000	0.0000	0.0000	0.0000	C.N.C.	C.N.C.	C.N.C.	C.N.C.	C.N.C.	C.N.C.	C.N.C.	C.N.C.	0.0000	C.N.C.
Kolmogorov–Smirnov test		Statistic(D)	0.2197	0.2243	0.2206	0.2335	0.1398	0.1593	0.1711	0.1434	0.1163	0.1173	0.1652	0.1503	0.1366	0.1734	0.1840	0.2254	0.1051	0.1769
		d.f.	14	14	14	14	14	14	14	14	14	14	14	14	14	14	14	14	14	14
		*p*	0.0655	0.0545	0.0632	0.0372	≥0.10	≥0.10	≥0.10	≥0.10	≥0.10	≥0.10	≥0.10	≥0.10	≥0.10	≥0.10	≥0.10	0.0521	≥0.10	≥0.10

MVC, maximum voluntary contraction; bEMG, background electromyograph; MEP, motor evoked potential; cSP, cortical silent period; rMT, resting motor threshold; SEM, standard error of the mean; NC, no-conditioned; C, conditioned; C.N.C., could not calculate.

**Table 2 brainsci-10-00063-t002:** Mean value and D’Agostino and Kolmogorov–Smirnov tests in the second experiment.

			Force[/MVC]										bEMG[/MVC]										MEParea[mV*ms]										cSP[ms]									
			NC	ISI0	ISI10	ISI20	ISI30	ISI40	ISI50	ISI60	ISI70	ISI80	NC	ISI0	ISI10	ISI20	ISI30	ISI40	ISI50	ISI60	ISI70	ISI80	NC	ISI0	ISI10	ISI20	ISI30	ISI40	ISI50	ISI60	ISI70	ISI80	NC	ISI0	ISI10	ISI20	ISI30	ISI40	ISI50	ISI60	ISI70	ISI80
Mean			0.18	0.18	0.17	0.17	0.18	0.17	0.17	0.17	0.18	0.18	0.27	0.27	0.26	0.28	0.27	0.28	0.30	0.25	0.27	0.28	0.81	0.84	0.78	0.76	0.71	0.71	0.71	0.74	0.76	0.78	102.5	103.1	92.2	88.4	85.5	84.8	88.7	92.7	93.0	91.8
SEM			0.02	0.02	0.02	0.02	0.02	0.02	0.02	0.02	0.02	0.02	0.04	0.04	0.03	0.04	0.03	0.04	0.04	0.03	0.04	0.04	0.14	0.15	0.13	0.14	0.15	0.14	0.13	0.14	0.14	0.15	7.1	7.2	6.7	6.5	7.3	7.4	7.4	7.2	7.2	6.1
D’Agostino test	Skewness	Statistic(Z)	2.52	2.35	2.4	2.32	2.35	2.39	2.32	2.3	2.3	2.41	1.2	1.92	0.35	1.15	0.59	1.95	1.57	1.22	1.39	0.69	1.25	1.31	1.36	1.68	1.80	1.19	0.83	1.77	1.58	1.65	1.81	1.17	1.58	1.86	2.26	1.58	1.93	1.69	1.55	1.93
		*p*	0.01	0.02	0.02	0.02	0.02	0.02	0.02	0.02	0.02	0.02	0.23	0.06	0.72	0.25	0.56	0.05	0.12	0.22	0.16	0.49	0.21	0.19	0.17	0.09	0.07	0.23	0.41	0.08	0.11	0.10	0.07	0.24	0.11	0.06	0.02	0.11	0.05	0.09	0.12	0.05
	Kurtosis	Statistic(Z)	C.N.C.	14.93	18.19	15.06	16.73	17	14.37	13.93	13.91	18.89	C.N.C.	C.N.C.	6.05	C.N.C.	5.82	C.N.C.	C.N.C.	C.N.C.	C.N.C.	5.34	C.N.C.	C.N.C.	C.N.C.	C.N.C.	C.N.C.	2.93	2.53	C.N.C.	C.N.C.	C.N.C.	C.N.C.	C.N.C.	C.N.C.	C.N.C.	C.N.C.	C.N.C.	C.N.C.	C.N.C.	C.N.C.	C.N.C.
		*p*	C.N.C.	0.00	0.00	0.00	0.00	0.00	0.00	0.00	0.00	0.00	C.N.C.	C.N.C.	0.00	C.N.C.	0.00	C.N.C.	C.N.C.	C.N.C.	C.N.C.	0.00	C.N.C.	C.N.C.	C.N.C.	C.N.C.	C.N.C.	0.00	0.01	C.N.C.	C.N.C.	C.N.C.	C.N.C.	C.N.C.	C.N.C.	C.N.C.	C.N.C.	C.N.C.	C.N.C.	C.N.C.	C.N.C.	C.N.C.
Kolmogorov–Smirnov test		Statistic(D)	0.21	0.21	0.22	0.21	0.2	0.23	0.22	0.2	0.21	0.22	0.14	0.15	0.17	0.15	0.12	0.18	0.14	0.11	0.19	0.11	0.15	0.18	0.16	0.16	0.20	0.22	0.21	0.16	0.15	0.17	0.18	0.15	0.16	0.14	0.21	0.15	0.15	0.22	0.14	0.23
		d.f.	14	14	14	14	14	14	14	14	14	14	14	14	14	14	14	14	14	14	14	14	14	14	14	14	14	14	14	14	14	14	14	14	14	14	14	14	14	14	14	14
		*p*	0.09	0.08	0.06	≥0.10	≥0.10	0.051	0.07	≥0.10	0.09	0.08	≥0.10	≥0.10	≥0.10	≥0.10	≥0.10	≥0.10	≥0.10	≥0.10	≥0.10	≥0.10	≥0.10	≥0.10	≥0.10	≥0.10	≥0.10	≥0.10	≥0.10	≥0.10	≥0.10	≥0.10	≥0.10	≥0.10	≥0.10	≥0.10	≥0.10	≥0.10	≥0.10	0.056	≥0.10	0.04

MVC, maximum voluntary contraction; bEMG, background electromyograph; cSP, cortical silent period; rMT, resting motor threshold; SEM, standard error of the mean; NC, no-conditioned; C.N.C., could not calculate.

## Data Availability

The datasets generated during and analyzed in the current study are available from the corresponding author upon reasonable request.

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
