# Peer review of "Cerebellar Transcranial Magnetic Stimulation Reduces the Silent Period on Hand Muscle Electromyography During Force Control"

_brainsci, 2020, doi:10.3390/brainsci10020063_

Round 1
Reviewer 1 Report
The study involves some interesting methodological work exploring cerebellar-M1 interaction. The two studies reported are sensible and well designed, but there are a number of omissions in the write up which need to be resolved. There are also formatting issues which prevent a full review of the statistics at present.
Method:
Figure 1: This is a minor and potentially rather picky point but is the handle of the TMS coil in the correct direction? For induced P-A current the coil is typically orientated the other way with the handle pointing away from the face.
Please include more details in the method. Specifically:
1: Were the participants were on any medications or have any current/ historic psychiatry conditions e.g. depression?
2: Where was your reference electrode located for EMG recording?
3: Does the TMS machine you use deliver mono or bi phasic pulses?
4: Please include a table showing the mean/SD MEP amplitudes and mean MSO for both regions.
5: What size coil was used for M1 stimulation? What size was the double cone coil?
6: How was the optimal site for stimulation of the FDI muscle identified? How was the coil location monitored and kept constant throughout the study?
7: What inter-trial-interval was used between pairs/ single TMS pulses?
8: Please define what bEMG stands for in the first use.
9: What software was used for statistical analysis?
As two independent assessors measured CSP offset it would be useful to see what sort of agreement was achieved before averaging. It would be useful to see something like: intraclass correlation coefficient or coefficient of variation values here.
Results:
Table 1 and 2 are unreadable. Please reformat so comments on this can be made.
Figure 3 and Figure 5 seem to be the wrong way around.
Please make sure all relevant figures are references in the text. This is not the case when reporting statistics for experiment 1.
Figure 3 caption which should really be attached to figure in fig 5 doesn’t adequately describe what figure d shows. Presumably this is conditioned/non conditioned.
Please note exact p values when reporting significant results using Scheffes test results for experiment 2.
Discussion:
Line 220 ‘these findings suggest that the cerebellum affected the contralateral M1 via GABAergic inhibitory neural circuits’. Without further explanation/rephrasing this is a bit out of context. The knowledge that GABAergic circuits are involved comes from past research. While it is likely that this does underlie the effects, this wasn’t really tested here.
Line 225 ‘may have long-term effects on cSP’. The use of ‘long term effects’ is a bit misleading here, consider rephrasing. At the moment it sounds like the authors are proposing that single pulse cerebellar stimulation leads to some sort of plasticity based change, but what I assume the authors are referring to is the <100ms CSP duration.
The effects of test pulse intensity over M1 are not discussed. A few sentences interpreting the results would be welcome.
Author Response
Response to reviewers’ comments
We greatly appreciate the helpful suggestions concerning our manuscript titled “Cerebellar transcranial magnetic stimulation reduces the silent period on hand muscle electromyography during force control.” We have addressed all the concerns of the reviewers and have carefully revised our manuscript. The changes are marked using track changes. Below are our point-by-point responses to the reviewers’ comments. Please let us know if further revisions are needed; we would be glad to incorporate them.
Summary of Revisions
Reviewer 1
(Comment #1)
Method:
Figure 1: This is a minor and potentially rather picky point but is the handle of the TMS coil in the correct direction? For induced P-A current the coil is typically orientated the other way with the handle pointing away from the face.
(Reply to #1)
Thank you for the important question. We applied the direction of the AP current in the coil to induce the PA travelling current in the brain (Sakai et al. 1997; Ni et al. 2011). We used a figure-of-8 coil (YM-132B, Nihon Kohden, Tokyo, Japan). We have reconfirmed the handle direction and found it to be correct.
(Comment #2)
Please include more details in the method. Specifically:
1: Were the participants were on any medications or have any current/ historic psychiatry conditions e.g. depression?
(Reply to #2)
Thank you for the very important question. There was no participant on medication and current/past history of psychiatric conditions. This explanation has been added to the Method section.
(Comment #3)
2: Where was your reference electrode located for EMG recording?
(Reply to #3)
Thank you for this question. The reference electrode was placed on the right dorsal wrist. This information has been added to the Method section.
(Comment #4)
3: Does the TMS machine you use deliver mono or bi phasic pulses?
(Reply to #4)
Thank you for this question. The magnetic stimulator (SMN-1200, Nihon Kohden, Tokyo, Japan) delivers a monophasic pulse. We have added this information to the Method section.
(Comment #5)
4: Please include a table showing the mean/SD MEP amplitudes and mean MSO for both regions.
(Reply to #5)
Thank you for the very important suggestion. The mean values, which were used for the figures, have been added to each table. The stimulation intensity was 90% MSO in all participant because CMEP did not appear, and this result is in line with the findings of some previous studies (Matsugi et al. 2013, 2014, 2015).
(Comment #6)
5: What size coil was used for M1 stimulation? What size was the double cone coil? (Reply to #6)
Thank you for this question. We used a figure-of-8 coil (YM-132B, Nihon Kohden, Tokyo, Japan) with an outer diameter of 99 mm and a double cone coil (YM-133B, Nihon Kohden, Tokyo) with an outer diameter of 60 mm. This information has been added to the Method section.
(Comment #7)
6: How was the optimal site for stimulation of the FDI muscle identified? How was the coil location monitored and kept constant throughout the study?
(Reply to #7)
Thank you for these questions. We located the hotspot, the largest MEP was obtained in left hemisphere on head of participant . Further, this hotspot was marked on the swimming cap the participant was wearing to prevent change in the stimulation site . This information has been added to the Method section.
(Comment #8)
7: What inter-trial-interval was used between pairs/ single TMS pulses?
(Reply to #8)
Thank you for the very important question. The inter-trial intervals were set to longer than 10 seconds. This information has been added to the Method section.
(Comment #9)
8: Please define what bEMG stands for in the first use.
(Reply to #9)
Thank you for this valuable suggestion. We have added the definition of bEMG.
(Comment #10)
As two independent assessors measured CSP offset it would be useful to see what sort of agreement was achieved before averaging. It would be useful to see something like: intraclass correlation coefficient or coefficient of variation values here.
(Reply to #10)
Thank you for the very important suggestion. We have added the analysis of the intraclass correlation coefficient (ICC) to the Method and Results sections.
(Comment #11)
Results:
Table 1 and 2 are unreadable. Please reformat so comments on this can be made.
(Reply to #11)
Thank you for noting this issue. We have revised the tables.
(Comment #12)
Figure 3 and Figure 5 seem to be the wrong way around.
(Reply to #12)
Thank you for this important point. We acknowledge this omission and have revised accordingly.
(Comment #13)
Please make sure all relevant figures are references in the text. This is not the case when reporting statistics for experiment 1.
(Reply to #13)
Thank you for this important point. We have rechecked the citation of the figures in the text and have revised accordingly.
(Comment #14)
Figure 3 caption which should really be attached to figure in fig 5 doesn’t adequately describe what figure d shows. Presumably this is conditioned/non conditioned.
(Reply to #14)
Thank you for noting this issue. We acknowledge this omission and have revised accordingly.
(Comment #15)
Please note exact p values when reporting significant results using Scheffes test results for experiment 2.
(Reply to #15)
Thank you for the very important suggestion. The p value was 0.00000000006, and this value has been reported in the Results section.
(Comment #16)
Discussion:
Line 220 ‘these findings suggest that the cerebellum affected the contralateral M1 via GABAergic inhibitory neural circuits’. Without further explanation/rephrasing this is a bit out of context. The knowledge that GABAergic circuits are involved comes from past research. While it is likely that this does underlie the effects, this wasn’t really tested here.
(Reply to #16)
Thank you for the very important suggestion. We have revised this sentence in the Discussion section.
(Comment #17)
Line 225 ‘may have long-term effects on cSP’. The use of ‘long term effects’ is a bit misleading here, consider rephrasing. At the moment it sounds like the authors are proposing that single pulse cerebellar stimulation leads to some sort of plasticity based change, but what I assume the authors are referring to is the <100ms CSP duration.
(Reply to #17)
Thank you for this valuable comment. We delivered single pulse TMS to the right cerebellum, but the period of cSP-modulation was 20 ms (from 20 ms to 40 ms after the conditioning stimulation). Therefore, we described the “long-term” effect on cSP. However, as the reviewer noted, this expression may lead to misunderstanding; therefore, we have revised the sentence.
(Comment #18)
The effects of test pulse intensity over M1 are not discussed. A few sentences interpreting the results would be welcome.
(Reply to #18)
Thank you for the very important suggestion. We have added one paragraph on the interpretation of the result of the test pulse intensity to the Discussion section.

Reviewer 2 Report
The rationale for the current study cites an EEG study (citation #15, line 59); however, this citation is to a review article on cerebellar non-invasive neuromodulation. The appropriate study citation is needed.
Figure 1 shows a coil generating a brain current traveling Anterior-Posterior according to citation #18; however, the text (line 101-102) says the current travels posterior-anterior. Clarification is needed. Also, the coil orientation in figure 1 is not conventional and a justification for the use of this coil direction should be added to the methods section.
Table 1 and 2: N.C. needs grammatical correction to "could not calculate"
The methods (line 113-115) state that if the cerebellar conditioning pulse did not evoke an MEP in FDI, that 90% MSO was used. It would be helpful for readers if it was stated in the results section how many participants did not have a cerebellar resting threshold, requiring 90% MSO for the cerebellar stimulation.
The order of the experiments is unclear. Was experiment 1 and 2 randomized or sequential between participants?
"bEMG" in the body text is not defined
The word "specimen" (line 198) is odd as this word is typically used to describe tissue or biologic samples. Perhaps the word "sample" or "example" could suffice nicely.
Figure 4 is very helpful for the reader to visualize the trend in CSP; however, the amplitude of the MEP is not easily observed. Adjusting the y-axis would allow the observation of the MEP or it should be stated in the text what the amplitude of the MEP was across ISI. This is vitally important because it is well established that cerebellar brain inhibition reduces MEP amplitude and this may be a factor in how the cSP is modulated with the ISI. Could the cSP be reduced because M1 excitability is reduced (reduction in MEP)?
Figure 5C: the comparison between 1rMT vs 1.3rMT reflects an effect of stimulation intensity presumably (higher stimulus generates a larger cSP); however, in line 217-218 the authors do not acknowledge that this is well established (see Kukowski and Haug 1992) and not a primary finding of the work. It should be stated if the authors think this is an intensity effect or the result of something else.
Discussion, line 219-220: this is a bold conclusion. One major limitation is the lack of analysis on the amplitude of the MEP. From the current analysis it is unclear if the MEP or M1 excitability is modulated at the ISI tested. If the MEP remains stable, there is support for the cerebellar conditioning to only act on GABAergic mechanisms; however, if the MEP is reduced at the optimal ISI of 20-40ms, that would suggest that M1 excitability (glutaminergic/excitatory) mechanisms may also be contributing to the reduction is cSP. The authors did not assess MEP amplitude, thus it is unclear if glutaminergic or other mechanisms can be ruled out.
Line 224-225: The authors state that cerebellar TMS may have long-term effects on cSP. However the study did not assess long-term, longitudinal or re-test time effects and thus, it is unclear what evidence they are referring to. Perhaps the authors meant effects on brain regions distal from the cerebellum?
A limitation of the discussion section is the lack of comment regarding the primary pathway from the cerebellum to M1. A comment regarding the disynaptic dento-thalamic-cortical pathway would strengthen the discussion in lines 226-231 (a good graphical source is Hahn & Paik, Brain & Neurorehabilitation, 2015).
Lines 232-237: This section discusses the clinical application of the findings but there are some conflicting statements. The authors state that cerebellar ataxia patients have no cerebellar-brain-inhibition (CBI) and that the cSP in prolonged, thus cerebellar output facilitates GABAergic inhibition. This line of thinking is confusing because the lack of CBI with co-existing prolonged cSP suggest that CBI actually reduces cSP. Because the absence of CBI (if facilitatory to GABA), it should result in reduced cSP (less GABA). This is actually what the results show, that stimulation to the cerebellum causes a reduction in cSP = reduced inhibition. Further, people with dystonia have been found to have no CBI response and have short cSP (Brighina, Exp Brain Res, 2009). Thus the interaction between CBI and cSP is very complicated and there is no direct evidence for the statement in lines 236-237. This is one reason this manuscript is actually very compelling for the field (this kind of work is very needed), but it does not prove cerebellar output to be simply "GABAergic". Perhaps I am misunderstanding the author's point though. I would encourage editing to clarify their position.
"SCD" in line 250 is not defined in the text
Author Response
Response to reviewers’ comments
We greatly appreciate the helpful suggestions concerning our manuscript titled “Cerebellar transcranial magnetic stimulation reduces the silent period on hand muscle electromyography during force control.” We have addressed all the concerns of the reviewers and have carefully revised our manuscript. The changes are marked using track changes. Below are our point-by-point responses to the reviewers’ comments. Please let us know if further revisions are needed; we would be glad to incorporate them.
Summary of Revisions
Reviewer 2
(Comment #1)
The rationale for the current study cites an EEG study (citation #15, line 59); however, this citation is to a review article on cerebellar non-invasive neuromodulation. The appropriate study citation is needed.
(Reply to #1)
Thank you for noting this issue. We have cited two original articles regarding this in the revised manuscript.
(Comment #2)
Figure 1 shows a coil generating a brain current traveling Anterior-Posterior according to citation #18; however, the text (line 101-102) says the current travels posterior-anterior. Clarification is needed. Also, the coil orientation in figure 1 is not conventional and a justification for the use of this coil direction should be added to the methods section.
(Reply to #2)
Thank you for noting this important issue. We used the AP current pulse in the coil, and this montage induce the PA current in the brain. This montage was used in our previous study (Matsugi, somatosensory and motor research 2019). We had explained this montage in “2.3. TMS over M1 and cerebellum.”
(Comment #3)
Table 1 and 2: N.C. needs grammatical correction to "could not calculate"
(Reply to #3)
Thank you for noting this. N.C. was converted from N.C. to C.N.C. (could not calculate).
(Comment #4)
The methods (line 113-115) state that if the cerebellar conditioning pulse did not evoke an MEP in FDI, that 90% MSO was used. It would be helpful for readers if it was stated in the results section how many participants did not have a cerebellar resting threshold, requiring 90% MSO for the cerebellar stimulation.
(Reply to #4)
Thank you for the very important suggestion. There was no participant with appearance of cervicomedullary MEP by cerebellar TMS with 100% MSO. Therefore, 90% MSO was applied in all examinations. Moreover, this result is in line with the findings of some previous studies (Matsugi et al. 2013, 2014, 2015). We have added the result regarding the CMEP.
(Comment #5)
The order of the experiments is unclear. Was experiment 1 and 2 randomized or sequential between participants?
(Reply to #5)
Thank you for this question. The first experiment was conducted first, and the second experiment was conducted second in all participants, i.e., there was no randomization.
(Comment #6)
"bEMG" in the body text is not defined
(Reply to #6)
Thank you for noting this. We have defined the term at first appearance in the revised manuscript.
(Comment #7)
The word "specimen" (line 198) is odd as this word is typically used to describe tissue or biologic samples. Perhaps the word "sample" or "example" could suffice nicely.
(Reply to #7)
Thank you for noting this. We have changed the word “specimen” to “sample” in the revised manuscript.
(Comment #8)
Figure 4 is very helpful for the reader to visualize the trend in CSP; however, the amplitude of the MEP is not easily observed. Adjusting the y-axis would allow the observation of the MEP or it should be stated in the text what the amplitude of the MEP was across ISI. This is vitally important because it is well established that cerebellar brain inhibition reduces MEP amplitude and this may be a factor in how the cSP is modulated with the ISI. Could the cSP be reduced because M1 excitability is reduced (reduction in MEP)?
(Reply to #8)
Thank you for the very important suggestion. In this study, the most important parameter is cSP; therefore, we adjusted the scale of the y axis for cSP. However, the MEP may affect the cSP. Therefore, we have added the analysis of MEParea, which reflects the excitability of the corticospinal tract during muscle contraction (Jono et al. 2016).
As a result, the MEP area at ISI 40-60 ms was significantly reduced, whereas cSP at ISI 20-40 ms was significantly reduced. If cSP reduction completely depended on the size of MEP, the MEP reduction would appear at ISI 20–40 ms. Therefore, we believe that the cSP reduction by cerebellar TMS may be nearly entirely independent from MEP modulation. However, we cannot rule out the possibility that cSP modulation depends on the size of the MEP at ISI 40 ms because only at this time the cSP and MEP areas were significantly reduced by cerebellar TMS. These sentences have been added to the Discussion section.
(Comment #9)
Figure 5C: the comparison between 1rMT vs 1.3rMT reflects an effect of stimulation intensity presumably (higher stimulus generates a larger cSP); however, in line 217-218 the authors do not acknowledge that this is well established (see Kukowski and Haug 1992) and not a primary finding of the work. It should be stated if the authors think this is an intensity effect or the result of something else.
(Reply to #9)
Thank you for this valuable comment. We acknowledge this previous finding that cSP depended on the intensity of TMS to M1. We have added this information to the Discussion section.
(Comment #10)
Discussion, line 219-220: this is a bold conclusion. One major limitation is the lack of analysis on the amplitude of the MEP. From the current analysis it is unclear if the MEP or M1 excitability is modulated at the ISI tested. If the MEP remains stable, there is support for the cerebellar conditioning to only act on GABAergic mechanisms; however, if the MEP is reduced at the optimal ISI of 20-40ms, that would suggest that M1 excitability (glutaminergic/excitatory) mechanisms may also be contributing to the reduction is cSP. The authors did not assess MEP amplitude, thus it is unclear if glutaminergic or other mechanisms can be ruled out.
(Reply to #10)
Thank you for the very important comment. We have added the analysis of MEP and discussion regarding this result.
(Comment #11)
Line 224-225: The authors state that cerebellar TMS may have long-term effects on cSP. However the study did not assess long-term, longitudinal or re-test time effects and thus, it is unclear what evidence they are referring to. Perhaps the authors meant effects on brain regions distal from the cerebellum?
(Reply to #11)
Thank you for this important question. We delivered single pulse TMS to the right cerebellum, but the period of cSP-modulation was 20 ms (from 20 ms to 40 ms after the conditioning stimulation). Therefore, we described the “long-term” effect on cSP. However, as the reviewer noted, this expression may lead to misunderstanding, and we have therefore revised this sentence.
(Comment #12)
A limitation of the discussion section is the lack of comment regarding the primary pathway from the cerebellum to M1. A comment regarding the disynaptic dento-thalamic-cortical pathway would strengthen the discussion in lines 226-231 (a good graphical source is Hahn & Paik, Brain & Neurorehabilitation, 2015).
(Reply to #12)
Thank you for the very important suggestion. We acknowledge that the dento-thalamic-cortical (DTC) pathway is involved in cerebellar brain inhibition (CBI) in the 5-8 ms test-condition stimulation interval, but CBI could not have appeared at > 10 ms (Ugawa et al. 1995). Therefore, we had excluded the DTC pathway from the possible pathways. However, as the reviewer notes, all the DTC and cortico-ponto-cerebellar and cortico-rubro-olivo-cerebellar pathways (Groiss and UGawa, 2011) are possible pathways involved in cSP-reduction by cerebellar TMS. We have added this important possible pathway to the Discussion section.
(Comment #13)
Lines 232-237: This section discusses the clinical application of the findings but there are some conflicting statements. The authors state that cerebellar ataxia patients have no cerebellar-brain-inhibition (CBI) and that the cSP in prolonged, thus cerebellar output facilitates GABAergic inhibition. This line of thinking is confusing because the lack of CBI with co-existing prolonged cSP suggest that CBI actually reduces cSP. Because the absence of CBI (if facilitatory to GABA), it should result in reduced cSP (less GABA). This is actually what the results show, that stimulation to the cerebellum causes a reduction in cSP = reduced inhibition. Further, people with dystonia have been found to have no CBI response and have short cSP (Brighina, Exp Brain Res, 2009). Thus the interaction between CBI and cSP is very complicated and there is no direct evidence for the statement in lines 236-237. This is one reason this manuscript is actually very compelling for the field (this kind of work is very needed), but it does not prove cerebellar output to be simply "GABAergic". Perhaps I am misunderstanding the author's point though. I would encourage editing to clarify their position.
(Reply to #13)
Thank you for the very important suggestion. As the reviewer notes, it is challenging to arrive to one conclusion regarding the mechanism of cSP reduction by cerebellar output. We also acknowledge that our expression may mislead the reader. Therefore, we have substantially revised this paragraph. Please let us know if further changes are needed.
(Comment #14)
"SCD" in line 250 is not defined in the text
(Reply to #14)
Thank you for the very important suggestion. The abbreviation “SCD” has been replaced with “spinocerebellar degeneration” in the Discussion section.

Round 2
Reviewer 1 Report
The authors have addressed my previous comments well and the addition of MEP analysis as suggested by the other reviewer is very informative. There are still a few issues which I would like to see resolved, but overall, I feel this is an interesting study which will be welcomed in the field.
Method: The authors have added information about the location of the reference electrode but then refer to a figure in which this isn’t shown. For consistency I suggest adding the reference to the figure.
Method: I find the reporting of the stimulation intensities a bit confusing. From page 4 lines 116 it seems that the cerebellar stimulation is set to 90% of the RMT measured from M1 pulses? But later in lines 121 it states that 90%MSO was used if 100% MSO did not yield an EMG response, and in the results section 90%MSO is reportedly used in ‘all examinations’. Please could the authors make it very clear how thresholds for each brain region were measured for each of the experiments? It would be helpful to be able to see the mean/SD of intensities used (as %MSO) across participants for M1, where I expect different intensities were used.
Results: page 9 lines 5 refers to result for cSP duration with reference to figure 3C, but this should be 3D? Please carefully check figure caption numbers.
Results: Figure 3c shows significant differences in MEP area when comparing unconditioned pulses at 1 and 1.3%RMT. This doesn’t appear to be mentioned in the text.
Results: Please check how figure 5 is referred to in the text. It seems 5C is referenced for cSP but this should be 5D.
Results: The results for analysis into MEP area shown in 5C are not reported in the text. These should be mentioned here or shown in a table. Interestingly, the differences appear to only be in relation to ISI of 0 (presumably CB and M1 pulse delivered at same time?) but not in comparison to unconditioned M1 pulses.
Table 2 is still unreadable (at least in the version I received to review). Please carefully check formatting.
General wording could be improved here: P3, lines 104: ‘The hotspot was defined as the site where the largest MEP was obtained in left hemisphere of head of participant’.
Author Response
Response to reviewers’ comments
We greatly appreciate the helpful suggestions concerning our manuscript titled “Cerebellar transcranial magnetic stimulation reduces the silent period on hand muscle electromyography during force control.” We have addressed all the concerns of the reviewers and have carefully revised our manuscript. The changes are marked using track changes. Below are our point-by-point responses to the reviewers’ comments. Please let us know if further revisions are needed; we would be glad to incorporate them.
Summary of Revisions
Reviewer 1
(Comment #1)
Method: The authors have added information about the location of the reference electrode but then refer to a figure in which this isn’t shown. For consistency I suggest adding the reference to the figure.
(Reply to #1)
Thank you for the suggestion. We added the reference electrode to the Figure 1.
(Comment #2)
Method: I find the reporting of the stimulation intensities a bit confusing. From page 4 lines 116 it seems that the cerebellar stimulation is set to 90% of the RMT measured from M1 pulses? But later in lines 121 it states that 90%MSO was used if 100% MSO did not yield an EMG response, and in the results section 90%MSO is reportedly used in ‘all examinations’. Please could the authors make it very clear how thresholds for each brain region were measured for each of the experiments? It would be helpful to be able to see the mean/SD of intensities used (as %MSO) across participants for M1, where I expect different intensities were used.
(Reply to #2)
Thank you for the comment. This RMT was measured from pulse to cerebellum (the coil was set to site 1cm below and 3 cm to the right of the inion). If cervicomedullary MEP was not obtain using 100%MSO, the 90%MSO was used for cerebellar TMS.
To avoid the confusion and increase readability, we revised these sentences and this description has been separated from the explanation of M1-TMS.
(Comment #3)
Results: page 9 lines 5 refers to result for cSP duration with reference to figure 3C, but this should be 3D? Please carefully check figure caption numbers.
(Reply to #3)
Thank you for the important suggestion. This is our mistake. we revised it.
(Comment #4)
Results: Figure 3c shows significant differences in MEP area when comparing unconditioned pulses at 1 and 1.3%RMT. This doesn’t appear to be mentioned in the text.
(Reply to #4)
Thank you for the important suggestion. This is our mistake. we added the result of analysis of MEP area.
(Comment #5)
Results: Please check how figure 5 is referred to in the text. It seems 5C is referenced for cSP but this should be 5D.
(Reply to #5)
Thank you for the suggestion. This is our mistake. We revised it.
(Comment #6)
Results: The results for analysis into MEP area shown in 5C are not reported in the text. These should be mentioned here or shown in a table. Interestingly, the differences appear to only be in relation to ISI of 0 (presumably CB and M1 pulse delivered at same time?) but not in comparison to unconditioned M1 pulses.
(Reply to #6)
Thank you for the important suggestion. We added the result of analysis of MEParea. In results, there was no significant difference between unconditioned trial and conditioned trials.
(Comment #7)
Table 2 is still unreadable (at least in the version I received to review). Please carefully check formatting.
(Reply to #7)
Thank you for the suggestion. We revised tables based on the guideline, and we can see all information of Table2. Just in case, we attached file as supplemental data of table2 for review.
(Comment #8)
General wording could be improved here: P3, lines 104: ‘The hotspot was defined as the site where the largest MEP was obtained in left hemisphere of head of participant’.
(Reply to #8)
Thank you for the suggestion. This sentence was revised, and also went through native check.

Reviewer 2 Report
Abstract: There is no mention of MEP results. Due to the significant effect of cerebellar conditioning on MEP size (clear evidence that M1 excitability is influenced by cerebellar conditioning at long ISI), it would be appropriate to incorporate.
Methods: Although the authors clarify why a AP current pulse was used, there is no rationale for the decision. From the citations used (#20-21) it is clear that PA and AP currents will produce different I-waves and different MEP latencies, thus it would be appropriate to state some rationale for choosing this coil orientation. Traditionally the opposite coil orientation is used for M1 stimulation (Rossini, Clin Neurophys, 2015) so it would be useful to know why the authors chose to use this non-traditional orientation. Is it easier to combine with cerebellar stimulation due to positioning of the coil handel or something else?
Results:
Figure 3C and 5C in text refers to the cSP data but Fig3/5C are actually the MEP data. I may have missed this detail, but I am unable to find what ISI was used to calculate the conditioned responses in Figure 3. It would be helpful to add this information to the legend because it appears the results will change if you choose the 40ms ISI (the one that reduced MEP and cSP values). There is no comment in the methods text regarding the MEP results.
Discussion:
Line 78: The authors state that dystonia is characterized by PROLONGED cSP and lack of CBI. This is not correct, dystonia is characterized by shortened cSP and absent CBI.Also, the Brighina 2009 paper did not assess cSP, only CBI. I cited this paper only to give the writers a reference for my point about CBI but it should not be used to make statements about cSP. My previous point and why I mentioned CBI in dystonia is that because both dystonia and cerebellar ataxia have absent CBI but opposite cSP abnormalities (ataxia has long cSP, dystonia has short cSP). Thus, drawing conclusions based on CBI or cSP from clinical populations is not well founded scientifically. Line 39: The word “shorted” should be corrected to "shortened" Line 46: The sentence “cSP shortening can indicate increased excitability of inhibitory neural circuits in M1” is confusing and does not agree with the literature (unless I misunderstand the writer). Typically it is accepted that cSP duration reflects inhibitory tone and longer cSP means more inhibition (Rossini, Clinical Neurophys, 2015). So cSP shortening reflects LESS inhibition or a DECREASE in excitability of inhibitory interneurons that medicate the cSP response. Thus the sentence in the discussion does not follow the accepted dogma of increasing cSP duration related to increasing GABAergic tone. The authors need to carefully re-state this section to clarify their position.
Author Response
Response to reviewers’ comments
We greatly appreciate the helpful suggestions concerning our manuscript titled “Cerebellar transcranial magnetic stimulation reduces the silent period on hand muscle electromyography during force control.” We have addressed all the concerns of the reviewers and have carefully revised our manuscript. The changes are marked using track changes. Below are our point-by-point responses to the reviewers’ comments. Please let us know if further revisions are needed; we would be glad to incorporate them.
Summary of Revisions
Reviewer 2
(Comment #1)
Abstract: There is no mention of MEP results. Due to the significant effect of cerebellar conditioning on MEP size (clear evidence that M1 excitability is influenced by cerebellar conditioning at long ISI), it would be appropriate to incorporate.
(Reply to #1)
Thank you for the important suggestion. We added the result of MEP to Abstract.
(Comment #2)
Methods: Although the authors clarify why a AP current pulse was used, there is no rationale for the decision. From the citations used (#20-21) it is clear that PA and AP currents will produce different I-waves and different MEP latencies, thus it would be appropriate to state some rationale for choosing this coil orientation. Traditionally the opposite coil orientation is used for M1 stimulation (Rossini, Clin Neurophys, 2015) so it would be useful to know why the authors chose to use this non-traditional orientation. Is it easier to combine with cerebellar stimulation due to positioning of the coil handel or something else?
(Reply to #2)
Thank you for the very important suggestion.
The current in the coil was directed from anterior to posterior, inducing a PA traveling current in the brain, and this current stimulation preferentially recruits the late I3-wave (Rothwell et al. 1991). Cerebellar TMS reduced electromyographical responses to I3 waves at an ISI of 6 ms, whereas it did not affect responses to I1 waves (Ugawa and Iwata 2005). This finding suggests that this cerebellar brain inhibition (CBI) is caused by suppression of I3 waves by cerebellar stimulation. Based on these findings, we thought that the PA direction current was suitable for observing the cerebellar TMS effect.
This explanation was added to method section.
(Comment #3)
Results:
Figure 3C and 5C in text refers to the cSP data but Fig3/5C are actually the MEP data. I may have missed this detail, but I am unable to find what ISI was used to calculate the conditioned responses in Figure 3. It would be helpful to add this information to the legend because it appears the results will change if you choose the 40ms ISI (the one that reduced MEP and cSP values). There is no comment in the methods text regarding the MEP results.
(Reply to #3)
Thank you for the important suggestion. These are our mistakes. We added the results of MEP and revised descriptions.
(Comment #4)
Discussion:
Line 78: The authors state that dystonia is characterized by PROLONGED cSP and lack of CBI. This is not correct, dystonia is characterized by shortened cSP and absent CBI.Also, the Brighina 2009 paper did not assess cSP, only CBI. I cited this paper only to give the writers a reference for my point about CBI but it should not be used to make statements about cSP. My previous point and why I mentioned CBI in dystonia is that because both dystonia and cerebellar ataxia have absent CBI but opposite cSP abnormalities (ataxia has long cSP, dystonia has short cSP). Thus, drawing conclusions based on CBI or cSP from clinical populations is not well founded scientifically.
(Reply to #4)
Thank you for very important suggestion. The first, this citation was mistake with misunderstand about Brighina 2009 paper. Therefore, we revised this citation.
In this paragraph, we did not want to state the relationship between CBI and cSP, but we wanted to describe our hypotheses about the reason of cSP-reduction by cerebellar TMS derived from our results and previous research. Our idea is not conflicting this reviewer’s opinion, and we understood reviewer’s idea. And we revised this paragraph based on above ideas.
(Comment #5)
Line 39: The word “shorted” should be corrected to "shortened"
(Reply to #5)
Thank you for the suggestion. we revised it.
(Comment #6)
Line 46: The sentence “cSP shortening can indicate increased excitability of inhibitory neural circuits in M1” is confusing and does not agree with the literature (unless I misunderstand the writer). Typically it is accepted that cSP duration reflects inhibitory tone and longer cSP means more inhibition (Rossini, Clinical Neurophys, 2015). So cSP shortening reflects LESS inhibition or a DECREASE in excitability of inhibitory interneurons that medicate the cSP response. Thus the sentence in the discussion does not follow the accepted dogma of increasing cSP duration related to increasing GABAergic tone. The authors need to carefully re-state this section to clarify their position.
(Reply to #6)
Thank you for the important question. “Typically it is accepted that cSP duration reflects inhibitory tone and longer cSP means more inhibition” is completely correct, and we understand so. Based on this interpretation, we described as “cSP shortening can indicate increased excitability of inhibitory neural circuits in M1.” However, this expression may have been confusing. We revised this description.
